# Study of Expression of *MST3* in Myeloid Leukaemia

**DOI:** 10.3390/medsci13020033

**Published:** 2025-04-01

**Authors:** Boro Arthi, Krishnaswamy Sujatha, Sridhar Gopal, Balasubramanian Balamuralikrishnan, Meyyazhagan Arun, Pappuswamy Manikantan, Palanisamy Sampathkumar, Arumugam Vijaya Anand

**Affiliations:** 1Department of Human Genetics and Molecular Biology, Bharathiar University, Coimbatore 641046, India; 2Department of Zoology, Government Arts College, Coimbatore 641018, India; 3Sri Ramakrishna Hospital, 395, Sarojini Naidu Road, Siddhapudur, Coimbatore 641044, India; 4Department of Food Science and Biotechnology, College of Life Science, Sejong University, Seoul 05006, Republic of Korea; 5Department of Life Sciences, CHRIST (Deemed to be University), Bangalore 560029, India; 6Department of Chemistry and Biosciences, SASTRA (Shanmugha Arts, Science, Technology & Research Academy) Deemed to be University, Srinivasa Ramanujan Centre, Kumbakonam 612001, India

**Keywords:** myeloid leukaemia, blood cell counting, expression, *MST3*, correlation

## Abstract

Myeloid leukaemia (ML) is a cancer that occurs by the accumulation of abnormally multiplied myeloid cells in bone marrow, peripheral blood, and other related tissue. *MST3* is a gene of the GCK family that has a role in apoptosis, along with other cellular functions like cellular differentiation, cell cycle, metabolism, and others. Objectives: The objectives of this study were to count RBCs and WBCs, study *MST3* expression in ML and control samples, and perform an in silico correlation study on the *KRAS* and *NRAS* genes. Methods: The counting of RBCs and WBCs was carried out using a hemacytometer, the expression of *MST3* was studied using RT-PCR, and a correlation study was carried out using GEPIA. Results: RBC and WBC levels in ML differed from the control levels, and the expression of *MST3* was found to be upregulated in ML in comparison to controls, with a 2.90–8.65-fold change, with a significant *p*-value > 0.05. A positive correlation in expression was also found between *MST3* and *KRAS* and *NRAS* genes, with a significant r value correlation. Conclusions: From this study, it could be deduced that *MST3* might have a role in ML pathogenesis, but further research is needed to study its role in the progression of the disease.

## 1. Introduction

Myeloid leukaemia (ML) is a severe blood-related disease. It is found to be prevalent in older people, but it occurs in children and younger adults as well. It is a heterogeneous disease that arises due to abnormal modification in hematopoietic stem cells, which causes proliferation, leading to the acquisition of mutations in the genes that cause failure in differentiation in these cells [1]. It is a disease of aberrant clonal expansion of the immature progenitors of myeloid cells and their accumulation in bone marrow and peripheral blood. It is an aggressive hematopoietic disease that causes high mortality. Most of the patients of this disease suffer from relapses due to the presence of heterogeneous clones [2]. It is a malignant malady in bone marrow and is a common form of acute leukaemia in older people with a diagnostic age of 65. The cause of this disease varies due to different factors such as patient age, low white blood corpuscle (WBC) count, mutations in genes like *NPM1*, *FLT3-ITD*, and *CEBPA*, and factors like chromosomal aberrations [3]. ML is a group of bone marrow diseases that occur in hematopoietic stem cells due to alterations in gene sequences that leads to dysregulated expression, causing myeloid sarcomas and leukaemia cutis. People suffering from this disease can be identified by routine blood work or by the presence of complicated symptoms like infections, bleeding, or intravascular coagulation [4]. The overall 5-year survival rate is 30%, and it differs in different age groups; it may reach 50% for patients of young ages and may be less than 10% in older patients [5]. Other than age and factors like sex, the progression of this disease is determined by the disease biology. Sequencing mutations in de novo genes, identified along with known mutations and recurrent mutations in *DNMT3A* and *IDH*, cause differences in clinicopathological features, in the progression of this disease, and affects the choice of treatment. Using novel molecular analysis like ultra-deep sequencing allows us to identify various genetic abnormalities that are included in schemes of risk stratification like National Comprehensive Cancer Network or European Leukemia Net (ELN) guidelines. Due to the discovery of mutations in the genes, the treatment option that used to be similar for all patients undergoing similar chemotherapy has expanded [6].

Germinal centre kinase (GCK) is a group of kinase proteins. These proteins are a subfamily of the sterile 20 kinase (Ste20) family and are involved in regulating several cellular functions and in determining the cell’s fate. There are different sub-groups of this protein family. GCK-1 is a mitogen-activated protein kinase *(MAPK*) that helps in activating the extracellular signal-regulated kinase that regulates cellular processes like cell proliferation and apoptosis. The GCK-II subfamily comprises the genes *MST-1* and *MST-2*, which are found to be abundant in lymphoid tissues. The interaction of *MST1* with *FOXO* is important for homeostasis in T-cells on the stimulus of apoptosis and is reported to play a critical role in controlling the proliferation of lymphocytes. In animal studies, *MST1* is found to have a role in chemotaxis in lymphocytes and thymocyte migration, is involved in the polarity of immune cells and adhesion, and controls their interstitial migration. The GCK-III includes the genes *MST3*, *MST4*, and *SOK1*, which are involved in apoptosis, cell adhesion, and polarity regulation [7]. Interactions of GCK-III subfamily genes with CCM3 have a role in the development of the cardiovascular system in zebrafish [8]. *MST3* is a kinase protein of the GCK family and a regulator for functions like growth, cell cycle, migration, and synapse development. Upregulation of the expression of *MST3* is involved in cell proliferation and migration in lung adenocarcinoma [9]. *MST3* in association with *STK26* regulates the reorientation of Golgi on the activation of RHO in cell migration; this protein also induces cell death by mediating oxidative stress by the phosphorylation of JNK1-JNK2 and p38 protein [10].

MST3, a serine/threonine kinase protein, has been explored in cancers like lung cancer and gastric cancer. A study carried out previously in paediatric ML indicated that the gene *MST3* may have a role in the pathogenesis of the disease, but this has not yet been properly explored [10]. Therefore, in the following study, the expression of *MST3* is carried out in patients with ML and controls to identify differentiation of *MST3* expression in ML samples. The objectives of this study include the quantification of red blood corpuscles (RBCs) and WBCs using a haemocytometer, the study of the expression of MST3 using real-time polymerase chain reaction (RT-PCR) analysis in ML and control samples, and an in silico correlation analysis of the expression of *MST3* with *KRAS* and *NRAS* in ML.

## 2. Materials and Methods

ML samples were collected from Sri Ramakrishna Hospital after obtaining ethical clearance from the Sri Ramakrishna Hospital Ethical Committee.

### 2.1. Counting of White Blood Corpuscles and Red Blood Corpuscles in Patients with Leukaemia 

The number of WBCs and RBCs were counted using a hemacytometer in the ML and control samples by diluting the samples in WBC and RBC diluting fluid using a WBC and RBC pipette. The diluted samples were then counted in the hemacytometer to obtain the total number of WBC and RBC cells/μL.

### 2.2. Expression of MST3 Using Real-Time Polymerase Chain Reaction

The expression of the gene *MST3* was carried out in 10 ML and 10 control samples using RT-PCR. Samples were collected from patients with ML who did not undergo any treatment, and consent was obtained. From both ML and control samples, mRNA was extracted using Trizol, Thermo Fisher Scientific Inc., Waltham, MA, USA. The mRNA obtained was further reverse transcribed into cDNA using a cDNA kit obtained from HiMedia, Mumbai, India. The expression of MST3 was carried out using MST3 forward primer 5′GGACTCAGAAAGTGGTTGCC3′ and reverse primer 5′AGCCTCCACCAAGATATTCCA3′ using SYBR Green q-PCR kit obtained from HiMedia, Mumbai, India in a q-PCR Applied Biosystems, Thermo Fisher Scientific Inc., Waltham, MA, USA.

### 2.3. Statistical Analysis

Statistical analysis was carried out using Student’s *t*-test to study the statistical inference of expression difference in *MST3* between the ML and control samples in SPSS software version 19 (IBM Corp., Armonk, NY, USA).

### 2.4. Correlation Analysis

An in silico correlation analysis of *MST3*’s expression with the gene *KRAS* and *NRAS* oncogenes belonging to the RAS family was carried out using the web-based tool GEPIA 2, Zhang Lab, Beijing, China in ML condition. The following analysis was carried out to identify the correlation of expressions of the genes in the ML data in TGCA and GTEx databases. Both the *KRAS* and *NRAS* genes were found to be related to ML.

## 3. Results

### 3.1. White Blood Corpuscle and Red Blood Corpuscle Counting

The number of WBCs and RBCs were found to be differentiated in the ML patient sample in comparison to controls. The number of WBCs was found to be high in ML samples of 18,000 cells/μL, and the number of RBCs was found to be low in ML samples of 3,930,000 cells/μL in comparison to the control levels, as listed in Table 1.

Description: Counting the number of WBCs and RBCs in ML and control samples, WBC levels in ML samples were found to be high in comparison to control levels, whereas RBC levels were found to be low in comparison to control levels.

### 3.2. Expression of MST3

*MST3* expression is upregulated in the ML samples compared to control samples, as shown in Table 2. The expression of *MST3* is illustrated in gel electrophoresis, and relative expression analysis is presented in Figure 1a,b. The expression of *MST3* is found to be differentiated in the ML samples in comparison to the control samples, with 2.90–8.65-fold change range.


medsci-13-00033-t002_Table 2Table 2Expression difference in the gene *MST3* in ML and control.SamplesExpression of *MST3*Expression (Fold Change)Control Sample1.33Sample 18.815.62Sample 210.977.25Sample 311.237.44Sample 410.16.59Sample 55.22.90Sample 611.047.3Sample 711.117.35Sample 812.838.65Sample 910.97.19Sample 1012.788.60


Description: RT-PCR expression of the target gene *MST3* in ML samples and fold change.


Figure 1(**a**) *MST3* expression in agarose gel electrophoresis with *GAPDH* as reference expressed gene. (**b**) RT-PCR expression analysis of *MST3*.
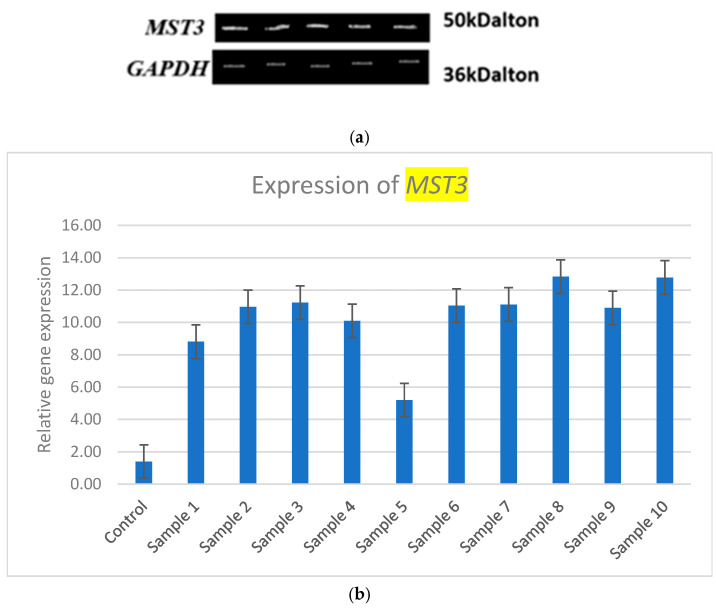



Description: Relative expression of the target gene *MST3* in controls and patients with ML.

### 3.3. Statistical Analysis of MST3 Expression

Student’s *t*-test was carried out to analyse the difference in the expression of the gene *MST3* between the control and ML samples. In the following analysis, the difference in expression of *MST3* between the control and ML samples is found to be significant, with a *p*-value of 0.0001 and with FDR of 0.05, which indicates the statistical significance of expression difference in MST3 between ML and control samples.

### 3.4. Correlation Analysis with MST3

According to in silico correlation analysis, the expression of the genes *KRAS* and *NRAS* and the expression of *MST3* were found to be correlated significantly, with significant r and *p*-values, as shown in Table 3 and Figure 2a,b.


medsci-13-00033-t003_Table 3Table 3Correlation of *KRAS* and *NRAS* with the gene *MST3* (also known as *STK24*).Genesr*p*-Value
*KRAS*
0.412.1 × 10^−8^
*NRAS*
0.484 × 10^−11^


Description: Correlation of the expression of *MST3* with the expression of the genes *KRAS* and *NRAS* was carried out in GEPIA; the correlation was found to have significant r and *p*-values.


Figure 2(**a**,**b**) Correlation of *KRAS* and *NRAS* with the gene *MST3* (also known as *STK24*).
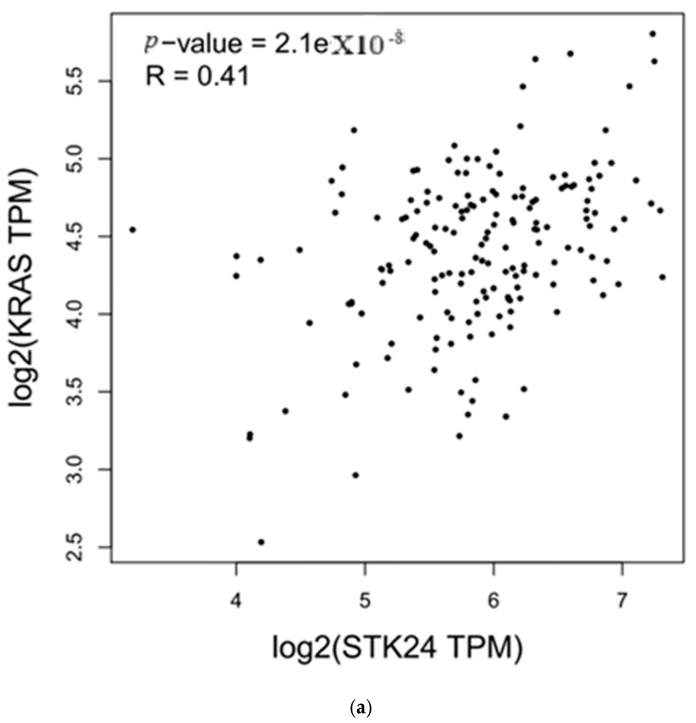

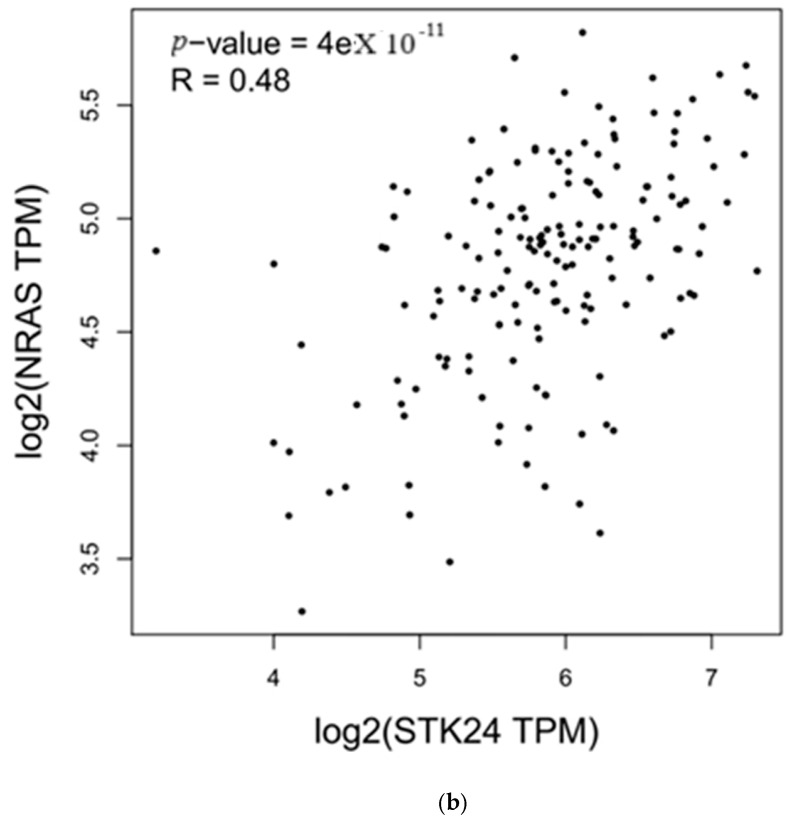



Description: *MST3* correlation with *KRAS* and *NRAS* in ML using GEPIA to identify the correlation of the genes’ expression in ML data in TGCA and GTEx databases.

## 4. Discussion

In the above study, the expression of the *MST3* gene with RBC and WBC counts were carried out in ML and control samples, and an in silico correlation analysis of *MST3* expression with the expression of the *KRAS* and *NRAS* genes was studied with the use of the web-based tool GEPIA. In this study, the expression of *MST3* was found to be upregulated in ML in comparison to control samples in RT-PCR expression analysis, with a fold change in the range of 2.90–8.65, with high WBC and low RBC counts in ML. In the following study, haemocytometers are used for counting RBCs and WBCs in ML and control samples. The following method was used, as it is a simple and accurate method for counting viable cells in samples and is also a reliable method for quantifying cellular components, giving a precise result. Using RT-PCR, the expression of the gene *MST3* was found to be upregulated in all ML samples compared to the control. The cause of difference in the upregulation of *MST3* expression in ML samples may be due to environmental conditions, epigenetic modifications, or mutational effects, but it needs further study to confirm the relationship of these conditions to *MST3* expression difference in ML and to validate *MST3*’s role in ML progression; further studies in in vitro models and animal models need to be carried out. Factors like epigenetic modification, mutation, and environmental factors contribute to alteration in gene expression, epigenetic modifications include nutritional components and toxins that contribute to the remodelling of the genome and modification of the expression of genes, genetic mutations lead to change in the sequence of the DNA that causes the alteration in gene expression, and environmental factors alter the expression by causing epigenetic modification, alteration in methylation, histone modification, and the influencing of transcription factors that repress or overexpress the genes as well. In Student’s *t*-test analysis, the upregulated expression of *MST3* in ML in comparison to controls was found to be significant, with a *p*-value > 0.05 (0.0001) and FDR of 0.05. This study was carried out on a small number of samples, so Student’s *t*-test was used for statistical analysis. In in silico correlation analysis, the *KRAS* and *NRAS* genes were found to have a significant correlation in expression with *MST3* under ML conditions having a correlation r of 0.41 and 0.48 and a *p*-value of 2.1 × 10^−8^ and 4 × 10^−11^. The correlation analysis in GEPIA was carried out to identify the expression correlation of the genes in the TGCA and GTEx databases of a particular cancer condition. In the following study, an analysis is carried out in which ML indicates a positive correlation between the *MST3* and *KRAS* and *NRAS* genes, but further correlations of *MST3* with *KRAS* and *NRAS* expression need to be performed in an animal model for validation. In the above study, the number of WBCs and RBCs were found to be differentiated from the control levels, which may be related to the difference in the expression of the *MST3* gene, but to conclude this, further animal studies need to be conducted.

The expression difference in genes in microarray studies was found to be related to the progression of ML. The genes that were found to be related were found to be overexpressed in ML in comparison to control samples [11]. Utilisation of the profiling of gene expression improves the classification of ML types, and analysis of gene expression is also a strong predictive prognostic factor for ML [12]. Serine/threonine kinase 39, a member of the Ste20 kinase family, plays a role in regulating the progression of tumours; silencing the expression of this kinase protein inhibits the proliferation of cells, promotes the differentiation of cells, induces apoptosis, and arrests the cell cycle [13]. Rho-associated coiled-coil kinase 2 (*ROCK2)*, a member of the serine/threonine protein kinase family, is found to have higher expression in patients with ML, which is related to a lower response rate, and elevation of the expression of this protein showed resistance to doxorubicin in the HL-60 cell line. Suppression of the expression of *ROCK2* enhances the sensitivity of the drug in ML [14].

*MST3* is a member of the mammalian Ste20 kinase family. In lung adenocarcinoma, the *MST3* gene was found to be over-expressed; in the cell line of A549 and H1299, the enhanced expression of *MST3* caused the proliferation and growth of the colony of both cell lines [9]. The expression of the *MST3* gene was found to be elevated in different tumour types and was found to have a role in phosphorylating the protein AKT at the amino acid position Thr21, which causes its activation and induces PD-L1, which clears its anti-tumour immunity modulator role and thereby shows it as a promising target for immunotherapy in cancer [15]. In non-small-cell-lung adenocarcinoma, the elevated expression of *MST3* is an independent prognostic factor. The alteration in the expression of the target gene is associated with a change in DNA amplification. The elevated expression of the following gene is related to unfavourable overall survival in non-small-cell-lung adenocarcinoma [16]. In comparison to normal lung tissues, the overexpression of *MST3* in lung cancer tissues is found to be positively related to a short survival time. High expression of the protein *MST3* is positively correlated with cancer proliferation, migration, invasion, and angiogenesis; silencing the expression of this gene obstructs tumour progression. Silencing the expression of *MST3* in an in vivo mouse model was found to regulate the progression of tumour formation and angiogenesis [17]. In gastric cancer, silencing the MST3 gene heightened the migration of the cells and increased metastasis. Expression of the *MST3* and *CDH1* genes was found to be correlated in gastric adenocarcinoma tissues, as a decrease in the expression of *MST3* suppressed the expression of *CDH1*, which increased the migration of the cells [18].

The diagnosis of ML can be carried out by calculating total blood cells like the number of WBCs and the number of RBCs [19]. A high WBC count was found to be related to the induction of mortality in ML patients and results in poor overall survival [20]. The status of WBCs and RBCs and haemoglobin evaluation reflects the condition of patients with ML after and before treatments, whether they improved or worsened [21]. Reduction in the number of WBCs may be associated with reduction in the alloimmunization of RBCs, which was found to be 2.8 % and 8.2% in non-WBC, reduced RBC alloimmunization [22]. The symptoms of leukopenia, anaemia, and pancytopenia were found to be more frequent in patients with ML than in ALL, which leads to the conclusion that haematopoiesis is inhibited more frequently in patients with ML than in ALL patients [23].

AI/ML is an emerging tool in the field of diagnosis of disease and can potentially be used in the screening and identification of ML cases. It is a relevant tool [24]. In the field of medicine, AI/ML is a capable tool that provides significant imaging aid and also generates a virtual cohort of individuals with a particular disease [25]. This field offers opportunities to advance the diagnosis of a disease by digitalizing images of a microscopic range. The conventional diagnosis of leukaemia is time-consuming, but by utilising AI, four types of leukaemia can be diagnosed with previous data [26]. This approach expands the capacity of humans to analyse large sets of complex data and provides a meaningful means of diagnosis, prognosis, and therapy for disease [27]. Advancement in algorithms in AI has improved in the prediction of disease progression, in optimising treatment response, and also in the stratification of patients according to different stages. Its use of genomic and epigenomic data has allowed for the discovery of novel molecular heterogenous insights into MDS and ML that lead to the development of new therapeutic strategies that are effective, and it allows for the development of tools for analysing the complex pattern of biopsy images of bone marrow for accurate diagnosis [28].

## 5. Conclusions

In this study, the expression of *MST3* was carried out in samples of patients with ML and controls, and *MST3* expression was found to be upregulated in ML in comparison to controls, with a significant *p*-value > 0.05 (0.0001). In the above conducted study, it can be indicated that the gene *MST3* has a role in ML. But further studies need to be carried out in in vitro models and in animal models to study the role of *MST3* in the progression of ML. 

In this study, an in silico correlation analysis of the expression of *MST3* with the *KRAS* and *NRAS* oncogenes of *RAS* family was carried out, and the correlation was found to be positive with significant r value correlation, but further validation of the correlation of these genes in ML in in vitro models and animal models needs to be performed. 

In the study, the numbers of RBCs and WBCs were altered, with a difference in the expression of *MST3*, but, to confirm, studies of ML in animal models need to be conducted to validate the correlation between the differential expression of *MST3* with RBC and WBC levels. From the above conducted study, it can be inferred that the *MST3* gene can be a positive target for the diagnosis and treatment of ML, but future research is needed to conclude further.

## Figures and Tables

**Table 1 medsci-13-00033-t001:** WBC and RBC counting in control and ML samples.

Samples	WBC (cells/μL)	RBC (cells/μL)
Control	5000	4,692,000
ML samples	18,000	3,930,000

## Data Availability

The data presented in this study are available on request to the researchers after approval from the corresponding author.

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
