# Peer review of "Study of Expression of MST3 in Myeloid Leukaemia"

_medsci, 2025, doi:10.3390/medsci13020033_

Round 1
Reviewer 1 Report
Comments and Suggestions for Authors
This study investigates the expression of MST3 in myeloid leukemia (ML) and its potential role in disease pathogenesis. While the study presents an interesting hypothesis, several methodological, analytical, and interpretational weaknesses must be addressed before publication.
- The study relies on a small cohort without independent validation. The sample size is not clearly justified, and there is no mention of power calculations. The authors should discuss the statistical power of their findings and ideally validate their results in an independent cohort or publicly available datasets.
- The methodology for blood cell counting is outdated. The use of a hemocytometer for RBC and WBC counting lacks precision compared to automated hematology analyzers. The authors should provide a justification for using this approach or discuss potential measurement biases.
- The study claims MST3 is upregulated in ML, but the results are inconsistent. Some ML samples show only marginal MST3 expression changes, while others display extreme fold changes. Were outliers removed or analyzed separately? The biological variability within the ML cohort needs further discussion.
- The statistical analysis is underdeveloped. The authors report a p-value of 0.0057, but there is no mention of correction for multiple comparisons. Given the small sample size, a false discovery rate (FDR) correction or Bonferroni adjustment should be considered.
- The discussion lacks a mechanistic framework. While the manuscript briefly mentions MST3's role in apoptosis and signaling, there is no mechanistic insight into how MST3 contributes to ML progression. Are there known MST3-interacting proteins involved in leukemia biology? Does MST3 impact key leukemogenic pathways such as FLT3, RAS, or PI3K/AKT?
- The study does not explore potential confounders. MST3 expression could be influenced by treatment status, patient age, or comorbidities. Were ML patients treatment-naïve? Were samples stratified based on disease subtypes (e.g., AML vs. CML)?
- There is no functional validation of the role of MST3 in ML. The study purely relies on expression analysis without testing whether MST3 knockdown or overexpression affects leukemic cell behavior. Functional experiments (e.g., siRNA knockdown, overexpression assays, cell proliferation/apoptosis assays) are needed to support the proposed role of MST3.
- The language requires significant revision for clarity and conciseness. For instance, “Expression of MST3 was found to be upregulated in myeloid leukemia than the control” should be revised to “MST3 expression was significantly upregulated in myeloid leukemia samples compared to controls.” Similarly, “The p-value was found to be > 0.05, indicating significance”.
- Figures require improvements. The gel electrophoresis image lacks molecular weight markers and proper loading controls. The RT-PCR bar graphs should include error bars and indicate statistical significance more clearly.
- The conclusion overstates the findings. While the study suggests MST3 may be relevant in ML, there is insufficient evidence to claim a direct pathogenic role. The manuscript should acknowledge that additional validation, mechanistic studies, and larger cohorts are needed.
Can be improved.
Author Response
Comment1:The study relies on a small cohort without independent validation. The sample size is not clearly justified, and there is no mention of power calculations. The authors should discuss the statistical power of their findings and ideally validate their results in an independent cohort or publicly available datasets.
Response 1: Thank you for the comment. The sample size of both ML and control are mentioned clearly and the statistical findings of expression difference are properly discussed with in the discussion section and are highlighted. As the expression in the control is found to have quite similar expression levels, therefore only one control expression data is taken.
Comment 2: The methodology for blood cell counting is outdated. The use of a hemocytometer for RBC and WBC counting lacks precision compared to automated hematology analyzers. The authors should provide a justification for using this approach or discuss potential measurement biases.
Response 2: Thank you for the comment. The reason for using hemocytometer for RBC and WBC counting is because it is the most reliable and accurate method for counting of the cells. The reason of choosing this method for counting RBC and WBC is been discussed briefly in the discussion section between 165-168. "In the following study, hemocytometer is used for counting RBC and WBC in ML and control samples, the following method is used as it is simple and accurate method for counting the viable cells in the samples and also a reliable method for quantifying the cellular components giving a precise result".
Comment 3: The study claims MST3 is upregulated in ML, but the results are inconsistent. Some ML samples show only marginal MST3 expression changes, while others display extreme fold changes. Were outliers removed or analyzed separately? The biological variability within the ML cohort needs further discussion.
Response 3: Thank you for the comment. The expression of MST3 in the ML samples are found to be upregulated from the control, changes are made which are highlighted in the Table 2. The outliers are removed. The variability in the expression of MST3 may be due to different conditions like mutation, epigenetic modification and environmental factors which are discussed briefly in the discussion section between 175-181 and are highlighted. "The factors like epigenetic modification, mutation and environmental factors contributes to alter the expression of the genes, epigenetic modifications includes nutritional components, toxins that contribute to remodeling of the genome and modifies the expression of genes, genetic mutations leads to change in the sequence of the DNA which causes the alteration of the expression of the genes, and environmental factors alters the expression by causing epigenetic modification, alteration of methylation and histone modification also influence transcription factors that repress or overexpress the genes"
Comment 4: The statistical analysis is underdeveloped. The authors report a p-value of 0.0057, but there is no mention of correction for multiple comparisons. Given the small sample size, a false discovery rate (FDR) correction or Bonferroni adjustment should be considered.
Response 4: Thank you for the comment. Due to the small sample size the statistical analysis is carried out using t-test. Certain changes are been made in statistical analysis with change in the expression data (Table 2) which are highlighted in the result section with FDR 0.05. " In the following statistical analysis, the difference in expression between the control and ML samples is found to be with a p-value of 0.0001, with FDR 0.05 which indicates the statistical significance of the difference in expression of MST3 in ML."
Comment 5: The discussion lacks a mechanistic framework. While the manuscript briefly mentions MST3's role in apoptosis and signaling, there is no mechanistic insight into how MST3 contributes to ML progression. Are there known MST3-interacting proteins involved in leukemia biology? Does MST3 impact key leukemogenic pathways such as FLT3, RAS, or PI3K/AKT?
Response 5: Thank you for the comment. A correlation analysis of the expression of KRAS and NRAS using in silico analysis tool GEPIA is carried out, in which the correlation is found to be significant. The following analysis is highlighted in the methodology, result and discussion section. Though the correlation is found to be significant in in silico further animal studies or in vitro studies are required to be done which is highlighted in the discussion section. The high expression of MST3 is found to induce the phosphorylation of AKT there by activates the protein which is highlighted. "In in silico correlation analysis the KRAS and NRAS which have role in pathogenesis in ML are found to have significant correlation in expression with MST3 in ML condition with r of .41 and .48 with p-value 2.1e-08 and 4e-11. The above correlation analysis is carried out using GEPIA which indicated a positive correlation in ML but further the correlation of MST3 with KRAS and NRAS expression are needed to be carried out in animal model to validate".
Comment 6: The study does not explore potential confounders. MST3 expression could be influenced by treatment status, patient age, or comorbidities. Were ML patients treatment-naïve? Were samples stratified based on disease subtypes (e.g., AML vs. CML)?
Response 6: Thank you for the comment. The samples of ML are collected from the patients with consent which didn't undergo any treatment. The samples are not stratified on disease subtypes, all the samples are collected from AML (ML) patients. "The samples of ML patients are collected from those who did not undergo any treatment with consent"
Comment 7: There is no functional validation of the role of MST3 in ML. The study purely relies on expression analysis without testing whether MST3 knockdown or overexpression affects leukemic cell behavior. Functional experiments (e.g., siRNA knockdown, overexpression assays, cell proliferation/apoptosis assays) are needed to support the proposed role of MST3.
Response 7: Thank you for the comment. In the following study the expression of MST3 is studied in ML samples for the first time. It was stated in a previous study that MST3 may have role in ML pathogenesis "In a study carried out previously in pediatric acute myeloid leukemia, the gene MST3 indicated to have a role in pathogenesis of the disease (10)", it is explored in ML for the first time. In vitro study of its role in ML progression is needed to be done in the future which is discussed briefly in the discussion section. In the following research we have only focused to study MST3's expression in ML samples but in future studies the role of MST3 in ML progression will be carried out further.
Comment 8: The language requires significant revision for clarity and conciseness. For instance, “Expression of MST3 was found to be upregulated in myeloid leukemia than the control” should be revised to “MST3 expression was significantly upregulated in myeloid leukemia samples compared to controls.” Similarly, “The p-value was found to be > 0.05, indicating significance”.
Response 8: Thank you for the comment. The above sentences are revised and written "In the following statistical analysis, the difference in expression between the control and ML samples is found to be with a p-value of 0.0001, with FDR 0.05 which indicates the statistical significance of the difference in expression of MST3 in ML".
Comment 9: Figures require improvements. The gel electrophoresis image lacks molecular weight markers and proper loading controls. The RT-PCR bar graphs should include error bars and indicate statistical significance more clearly.
Response 9: Thank you for the comment. The molecular weight marker is added in the gel electrophoresis image with loading control GAPDH. In RT-PCR bar graph the error bars have been added. The statistical significance of expression of MST3 is carried out clearly using student's t-test.
Comment 10: The conclusion overstates the findings. While the study suggests MST3 may be relevant in ML, there is insufficient evidence to claim a direct pathogenic role. The manuscript should acknowledge that additional validation, mechanistic studies, and larger cohorts are needed.
Response 10: Thank you for the comment. The conclusion of the following study states that MST3 has role in ML but to confirm its role in progression further in vitro and animal studies are required which are added and highlighted in the conclusion section. "In the following study, the expression of MST3 is carried out in the ML patient samples with controls, in which MST3 expression is found to be dysregulated in ML in comparison to controls which is found statistically significant with a p-value > 0.05. From the above conducted study, it can be surmised that the gene MST3 has a role in ML, but further MST3’s role in ML is needed to be validated by in vitro and animal studies and also further study is needed to carry out MST3’s role in progression of ML. Study of expression of MST3 in a large number sample of ML is also needed to be done further."

Reviewer 2 Report
Comments and Suggestions for Authors
The paper is generally well writen and wihin the scope of the journal. It is generally interesting and has potential to be published. However, the paper needs significant improvements in order to reach standard quality level of a journal paper. Here are the suggestions to be considered by the authors:
- Novelty of the paper appears critical, therefore it has to be clearly expressed by the authors themselves. The last paragraph of the Introduction needs to be expanded expressing briefly contribution and novelty of the study, beyond alredy clearly expressed objective of the study.
- Conclusion section needs improvements and expansion. The conclusions need to be expressed more directly and more elaborated and connected with results. Also, besides stements on the need for further verification of the main conclusion, short ellaboration on the further research directions is advisable to be included.
- Due to the current trend of wide application of AI/ML approaches, authors would enhance interest and support citation probability for the paper if they include short remark, in light of the previous point, about possible application of AI/ML regarding main objective of the study. Does the main objective allows for improved models, diagnosis, personalized treatments pof ML?
- Abbreviations in abstract and keywords are too extensive and should generally be avoided, especially as keywords. In abstract no matetr how well knwn are the abeviations, they need to be defined with the first occurence or even better replaced with full meaning avoiding repeated occurrence.
- The paper appears a bit scetchy and unfinished. Especially the Section Materials and Methods could be expanded, along with remarks directed towards improvenets of the Introduction and Conclusions sections.
- Figure 1(a) appears not to be fully visible.
- In Figure 1(b) each group is marked twice in the graph, both in the graph itself and in the legend, which appears confusing. Please clarify.
The English is generally OK and if and when the paper reches publication stage only the final proofreading might be needed.
Author Response
Comment: Novelty of the paper appears critical, therefore it has to be clearly expressed by the authors themselves. The last paragraph of the Introduction needs to be expanded expressing briefly contribution and novelty of the study, beyond already clearly expressed objective of the study.
Response: Thank you for the comment. The differential expression in genes has been found to be related with ML condition. MST3 gene has been explored in the cancer of lung and gastric but it has not been explored in ML. Therefore in the introduction section details regarding the study has been added and highlighted. "MST3, a serine/threonine kinase protein has been explored in cancer like lung cancer and gastric cancer. A study carried out previously in pediatric ML in which the gene MST3 indicated to may have a role in pathogenesis of the disease but has not been explored properly yet (10). Therefore, in the following study, the expression of MST3 is carried out in ML patients and controls to identify differentiation of MST3 expression in ML samples ".
Comment: Conclusion section needs improvements and expansion. The conclusions need to be expressed more directly and more elaborated and connected with results. Also, besides stements on the need for further verification of the main conclusion, short ellaboration on the further research directions is advisable to be included.
Response: Thank you for the comment. The conclusion section has been expanded by addition of future studies details that is needed to be done "In the above conducted study, it can be indicated that the gene MST3 has a role in ML. But further studies need to be carried out in in vitro models and animal models to study the role of MST3 in progression of ML. In this study an in-silico correlation analysis of the expression of MST3 with KRAS and NRAS oncogenes of RAS family is carried out, in which the correlation is found to be positive with significant correlation r value, but further validation of correlation of these genes in ML condition in in vitro and animal models needed to be done. In the study the numbers of RBC and WBCs are altered with difference of expression of MST3, but to confirm, studies in animal models need to be done to validate the correlation between the differential expression of MST3 with RBC and WBC levels in ML. From the above conducted study it can be inferred that MST3 gene can be a positive target for diagnosis and treatment of ML, but future research is needed to conclude further".
Comment: Due to the current trend of wide application of AI/ML approaches, authors would enhance interest and support citation probability for the paper if they include short remark, in light of the previous point, about possible application of AI/ML regarding main objective of the study. Does the main objective allows for improved models, diagnosis, personalized treatments pof ML?
Response: Thank you for the comment. AI/ML is a novel approach which is used in diagnosis and prediction of effective treatment for ML disease which has been added in the discussion section and are highlighted. "AI/ML is an emerging tool in the field of diagnosis of disease and can potentially be used in screening and identification of ML cases. It is a relevant tool (25). In the field of medicine AI/ML is a capable discipline that provides significant to the imaging, also generate virtual cohort of individuals of a particular disease (26). This field offers opportunities to advance the diagnosis of diseases, by digitalizing the image of microscopic range. Conventional diagnosis of leukemia is time-consuming but utilizing the AI four types of leukemia are diagnosed with the data (27). This approach expands the capacity of humans to analyze large set of complex data and provide meaningful tool for diagnosis, prognosis and therapy of the diseases (28). Advancement in algorithms in AI has improved in prediction of the disease progression, in optimizing the treatment response and also in stratification of the patients according to different stages. Its use of genomic and epigenomic data has discovered novel molecular heterogenous insight in the MDS and ML that lead to development of therapeutic strategies that are effective and develops analyzing tool for analyzing the complex pattern of biopsy image of bone marrow for potential and accurate diagnosis (29)".
Comment: Abbreviations in abstract and keywords are too extensive and should generally be avoided, especially as keywords. In abstract no matetr how well knwn are the abeviations, they need to be defined with the first occurence or even better replaced with full meaning avoiding repeated occurrence.
Response: Thank you for the comment. The abbreviated keywords are removed from keyword and changes are also made in the abstract.
Comment: The paper appears a bit scetchy and unfinished. Especially the Section Materials and Methods could be expanded, along with remarks directed towards improvenets of the Introduction and Conclusions sections.
Response: Thank you for the comment. The methods and methodology section is improved by adding the sample numbers of both ML and control and also by addition of more details regarding the methodology.
Materials and Methods
The ML samples are collected from Sri Ramakrishna Hospital after obtaining ethical clearance from Sri Ramakrishna Hospital Ethical Committee.
2.1 Counting of white blood corpuscles and red blood corpuscles in leukemia patients.
The number of WBC and RBC are counted using the hemocytometer in the ML and control samples by diluting the samples in WBC and RBC diluting fluid using WBC and RBC pipette. The diluted samples are then counted in the hemocytometer to obtain the total number of WBC and RBC cells/ μl.
2.2 Expression of MST3 using Real-time polymerase chain reaction.
The expression of the gene MST3 is carried out in 10 ML and 10 control samples using RT-PCR. The samples of ML patients are collected from those who did not undergo any treatment with consent. From both ML and control samples mRNA is extracted using Trizol. The mRNA obtained is further reversed transcribed into cDNA using a cDNA kit obtained from HiMedia. The expression of MST3 is carried out using MST3 forward primer 5'GGACTCAGAAAGTGGTTGCC3' and reverse primer 5'AGCCTCCACCAAGATATTCCA3' using SYBR Green q-PCR kit obtained from HiMedia in a q-PCR.
2.3 Statistical analysis
Statistical analysis is carried out using student’s t-test to study the statistical inference of expression difference of MST3 between the ML and control samples in SPSS software.
2.4 Correlation analysis
An in-silico correlation analysis of the MST3’s expression with the gene KRAS and NRAS oncogenes belonging to RAS family is carried out using the web-based tool GEPIA in ML condition. The following analysis is carried out to identify the correlation of expression of the genes in the data of ML from TGCA and GTEx database. Both the genes KRAS and NRAS are found to be related to ML.
The conclusion section too is improved by adding more details about the future works that are needed to be done.
Conclusions
In the following study, the expression of MST3 is carried out in the ML patient samples with controls, in which MST3 expression is found to be upregulated in ML in comparison to controls with significant p-value > 0.05 (0.0001). In the above conducted study, it can be indicated that the gene MST3 has a role in ML. But further studies need to be carried out in in vitro models and animal models to study the role of MST3 in ML progression. In this study an in-silico correlation analysis with expression of MST3 with KRAS and NRAS oncogenes of RAS family is carried out, in which the correlation is found to be positive with significant correlation r value, but further validation of correlation of these genes in ML condition in in vitro models and animal models needed to be done. In the following study the numbers of RBC and WBCs are altered with difference of expression of MST3, but to confirm, studies in animal models need to be done to validate the correlation between the differential expression of MST3 with RBC and WBC levels in ML. From the above study it can be inferred that MST3 gene can be a positive target for diagnosis and treatment of ML, but future research is needed to conclude further.
Comment: Figure 1(a) appears not to be fully visible.
Response: Thank you for the comment, the figure has been changed with addition of the GAPDH as reference expressed gene.
Comment: In Figure 1(b) each group is marked twice in the graph, both in the graph itself and in the legend, which appears confusing. Please clarify.
Response: Thank you for the comment, the figure 1(b) has been changed with addition of error bars in the graph of MST3 expression in ML.

Round 2
Reviewer 1 Report
Comments and Suggestions for Authors
The authors have provided detailed and well-structured responses to the issues I raised. I believe the manuscript is now suitable for publication.
Comments on the Quality of English LanguageCan be improved.
Reviewer 2 Report
Comments and Suggestions for Authors
I acknowledge effort that has been invested by the authors in the paper improvements. All my concerns have been thoroughly and meaningfully addressed, which I believe has resulted in a significantly beter version of the paper, which retained the essence of the study but reached the quality level expected from the paper to be published in Medical Sciences journal.
My final minor remark is that current expanded Conclusions section would be made more readible if it was divided into two or more subparagraphs. Simply start new line for sentences starting with words 'In' in the newly added text in Conclusions section.
Apart from this minor suggestion, with pleasure I support publishing of the current version of the paper.